Description of the new species Sigambra nkossa (Annelida, Pilargidae), with an analysis of the distribution patterns of polychaetes associated with artificially hydrocarbon-enriched bottoms

Martin Daniel dani@ceab.csic.es 1
Gil João 2
Chaineau Claude-Henri 3
Thorin Sébastien 4
Le Gall Romain 4
Dutrieux Eric 4
1 Center for Advanced Studies of Blanes, Consejo Superior de Investigaciones Científicas , Blanes , Catalunya , Spain
2 Centre of Marine Sciences, CCMAR, University of Algarve , Faro , Portugal
3 Department of HSE/EP/Environment, TotalEnergies , Paris , La Défense Cedex , France
4 CREOCEAN , Montpellier , France
Magalhães Wagner
Electronic publication date: 2022 Oct 19
Publication date: 2022
Volume: 10
Electronic Location ID: e13942
Received 2022 Jan 31; Accepted 2022 Aug 3
Copyright: ©2022 Martin et al.
Copyright year: 2022
Copyright holder: Martin et al.
License: This is an open access article distributed under the terms of the Creative Commons Attribution License, which permits unrestricted use, distribution, reproduction and adaptation in any medium and for any purpose provided that it is properly attributed. For attribution, the original author(s), title, publication source (PeerJ) and either DOI or URL of the article must be cited.
License URL: https://creativecommons.org/licenses/by/4.0/

Keywords: Hydrocarbon enriched sediments, Benthic fauna, Polychaetes, Pilargids, New species, Gulf of Guinea

Funding: Consolidated Research Group on Marine Benthic Ecology of the Generalitat de Catalunya 2017SGR378 CSIC Intramural Project 201630E020, and the ongoing project “Study of natural systems affected by coastal management and infrastructure projects in the open sea” involving the CEAB-CSIC and CREOCEAN CREOCEAN CSIC Open Access Publication Support Initiative from the CSIC Unit of Information Resources for Research (URICI) This article was funded by support given to Daniel Martin by the Consolidated Research Group on Marine Benthic Ecology of the Generalitat de Catalunya (Ref. 2017SGR378), the CSIC Intramural Project 201630E020, and the ongoing project “Study of natural systems affected by coastal management and infrastructure projects in the open sea” involving the CEAB-CSIC and CREOCEAN. João Gil was funded by a collaborative agreement signed with CREOCEAN. Daniel Martin received funds supporting the payment of the PeerJ publication fees through the CSIC Open Access Publication Support Initiative from the CSIC Unit of Information Resources for Research (URICI). TotalEnergies E&P Congo partly sponsored the field surveys. Staff members of CREOCEAN participated in the study design and in sample collection cruises.

==============================
The monitoring of the N’Kossa offshore oil and gas fields in the Republic of Congo allowed us to assess the ecological traits of two polychaete species belonging to Sigambra (Annelida, Pilargidae). Sigambra parva occur in very low densities in all bottoms, except the most impacted, where it is totally absent; it is an undescribed species that reached >4,000 ind/m2 in hydrocarbon-enriched sediments. Their distribution patterns are compared with those of other polychaetes, showing a range of affinities for hydrocarbon-enriched sediments in the N’Kossa region. Our results suggest that S. parva would be a representative of the original local fauna, while the species associated with artificial hydrocarbon-enriched sediments, including the other Sigambra, six more polychaetes and a bivalve, could be natively associated with natural hydrocarbon-enriched sediments, using the former as alternative habitats and as dispersal stepping stones. This ecological segregation, together with a careful morphological and morphometric analyses led us to describe the latter as a new species, namely Sigambra nkossa sp. nov. Moreover, morphometric analysis allowed us to discuss on the taxonomic robustness of the key morphological characters of S. nkossa sp. nov., as well as to emend the generic diagnosis of Sigambra to accommodate the new species.

Introduction

The structure of the meio- and macroinfaunal assemblages has been often used as a bioindicator to assess the environmental impact of both crude oil spills (e.g., Gómez Gesteira & Dauvin, 2005; Washburn, Rhodes & Montagna, 2016) and offshore hydrocarbon (HYD) extraction activities (e.g., Montagna & Harper, 1996; Carroll et al., 2000). The drill mud disposal in offshore oil and gas fields off the Congolese coasts of the Gulf of Guinea (Western Africa) is not an exception (see Denoyelle et al., 2010, and references therein). HYDs are an important component of drilling muds that often accumulate in the bottoms surrounding offshore oil and gas platforms, leading to the presence of highly modified macroinfaunal assemblages.

Alterations caused by HYD accumulation may occur both at assemblage and individual levels and may include decreasing abundances and diversities or reduced growth and reproductive rates, respectively (e.g., Olsen et al., 2007; Washburn, Rhodes & Montagna, 2016 and references therein). The macroinfaunal patterns in the vicinity of drilling platforms acting as local sources of HYD enriched sediments (HES) often do not differ from those related to other sources of organic enrichment. It is thus possible to find the typical succession mirroring an organic enrichment gradient, ranging from more diverse assemblages with low abundant, sensitive species to low diverse assemblages with highly abundant, opportunistic species, while the most enriched (i.e., polluted) areas are devoid of infauna in extreme cases (Pearson & Rosenberg, 1978). The main difference in the succession patterns affects the indicator species associated with the source of organic enrichment, which in this case, are obviously HYD tolerant (Montagna & Harper, 1996; Dalmazzone et al., 2004; Denoyelle et al., 2010).

Private companies are responsible for many worldwide activities related with offshore HYD routine exploration and extraction. Thus, access to these highly interesting macrofaunal and environmental data is often only possible through environmental assessment or monitoring projects. These projects are frequently a legal imperative to companies exploiting natural resources in the sea and habitually include benthic monitoring. Thus, they regularly generate large macrobenthic datasets, sometimes including reference invertebrate collections and interesting information from often previously poorly known or non-studied areas. This can be seen in a recent study carried out in the Falkland Islands, where only 16% of the 191 species found were known species described from the region, while 13% were uncertain (i.e., damaged, presumed cosmopolitan or having poor original descriptions) and the remaining ones (almost 70%) were unknown, likely new (Neal et al., 2020). A similar situation occurs with the surveys carried out in association to the oil and gas activities of TotalEnergies in the Atlantic coasts of the Gulf of Guinea facing the Republic of Congo. The datasets of benthic macrofauna generated have been partly managed through the long-term collaboration between the scientists of the Centre d’Estudis Avançats de Blanes (CEAB-CSIC) and the French company CREOCEAN. The in-depth analysis of these datasets proved to have a high scientific and management interest, including studies on sediment toxicity and recolonization (Dalmazzone et al., 2004) or on the contrasted role as indicators of foraminifers vs. macrofauna (Denoyelle et al., 2010).

Framed within the monitoring studies sponsored by TotalEnergies E&P Congo around the N’Kossa field off the Republic of Congo, we are here analyzing the influence of the environmental characteristics of the area on the distribution of several polychaete species showing a range of affinities for HES, which were found during the monitoring surveys carried out by CREOCEAN and the CEAB-CSIC. Based on previous results on the macrofauna of the area (Dalmazzone et al., 2004; Denoyelle et al., 2010), the species selected to be studied in deep include Capitella sp. (Capitellidae), Raricirrus sp. (Ctenodrilidae), Paramphinome trionyx Intes & Le Loeuff, 1975 (Amphinomidae), Oxydromus berrisfordi (Day, 1967) (Hesionidae), Lindaspio sebastiena Bellan, Dauvin & Laubier, 2003 (Spionidae), an unidentified species of Ampharetidae (referred to as Ampharetidae sp.) and two species of Sigambra (Pilargidae), Sigambra parva (Day, 1963) and an undescribed species (Day, 1963; Day, 1967; Intes & Le Loeuff, 1975; Bellan, Dauvin & Laubier, 2003). The implications of the presence of HES-associated species in artificially enriched, off-shore, shallow-water soft bottoms are discussed in light of the currently available knowledge on the ecological relationships between polychaetes and HYD.

We would also like to stress the importance of the studies focusing on taxonomy, which constitute an excellent example of the potential contribution of monitoring surveys to increase the current knowledge on marine biodiversity. Accordingly, several new species have been described from areas of the Congolese Oil & Gas licenses exploited by TotalEnergies E&P Congo. These include the vesicomyid bivalve Isorropodon bigoti Cosel, & Salas, 2001 and two polychaete annelids, L. sebastiena and Anotochaetonoe michelbhaudi Britayev & Martin, 2006 (Polynoidae) (Cosel & Salas, 2001; Bellan, Dauvin & Laubier, 2003; Britayev & Martin, 2006). To date, the first two species are known only from their type localities, which are the HES bottoms just below the extraction platforms of the N’Kossa field, while the third one was also reported from muddy bottoms in the Gulf of Cádiz, in an area rich in mud volcanoes and pockmarks (Ravara et al., 2017).

In the present article, we are reporting a third new species of polychaete for the N’Kossa field, which belongs to the genus Sigambra (Pilargidae). Together with its formal description, we are comparing the new species with all previously known species of Sigambra, we are performing a morphometric analysis of the relationships between size and several taxonomically relevant features, which allows us to discuss on its taxonomic robustness, and we are amending the generic diagnosis of Sigambra to accommodate the new species.

Materials & Methods

Sample collection, treatment and statistical analyses

During regular environmental monitoring surveys approved by TotalEnergies E&P Congo, samples for macrofaunal analyses were collected around the NKF I and NKF II platforms in the N’Kossa oil and gas field (180 m depth, 60 km off the coast of the Republic of Congo, Gulf of Guinea, Atlantic coasts of western Africa) (Fig. 1) as described in Britayev & Martin (2006) and Denoyelle et al. (2010). Specifically, stations 2, 3, 4 and 6 were sampled in November 2000, March 2002 and April 2003; stations 1, 5 and 7–10 only in 2000; stations 13 and 16 in 2002 and 2003; station 15 only in 2002; and stations 17–20 only in 2003. All specimens of the eight target polychaete species were identified, counted and their densities expressed as number of individuals per m2 (ind/m2, Table S1).

Figure 1 Geographical location of the study site and sampling stations.

(A) Map of Africa showing the location of the Republic of Congo. (B) Map of the Republic of Congo showing the location of the N’Kossa field. (C) Location of the drilling points at NKF 1 and NKF2 and sampling stations in the N’Kossa field.

The stations were environmentally characterized by the distance from drilling point (DDP) expressed in meters (m), as well as by a series of physical-chemical sediment descriptors (Table S1), which were analyzed in the Laboratoire Municipal et Régional de la Ville de Rouen (France) as described in Dalmazzone et al. (2004).

The possible existence of temporal and spatial differences in sediment descriptors, as well as in the presence of the target species was assessed by Parametric MultiDimensional Analysis (PMDA), based on the Mahalanobis distance and applying the Bonferroni correction to prevent Type I errors (Bland & Altman, 1995). The influence of sediment descriptors on the studied samples was analyzed by principal component analysis (PCA). Both types of data, sediment descriptors and polychaete densities, were plotted on the obtained axis to graphically visualize their distributions. The relationships between species densities and sediment descriptors were assessed by Pearson correlation. The significance of the differences between the sample groups obtained in the PCA was assessed by analysis of similarity (ANOSIM) and confirmed by PMDA (same procedure as above), both for sediment descriptors and target species. The significance of the differences for each sediment descriptor and target species was then assessed by independent one-way Analysis of Variance (one-way ANOVA), and the group(s) responsible for the observed differences were assessed by a post hoc Tukey test. All analyses were based on log-transformed data to meet with the assumptions of normality and homoscedasticity required for parametric analyses (Zar, 1984) and were performed with the XLSTAT software, version 2016.02.27390 (copyright Addinsoft 1995–2016), except the PCA and ANOSIM, which were performed with the respective routines of the Primer 6 software, version 6.1.11, and PERMANOVA, version 1.0.1 (copyright Primer-e Ltd. 2008). Other abbreviations used in the text are: Ba, barium; CS, coarse sand; DDP, distance from drilling point; DW, dry weight; HES, hydrocarbon enriched sediments; HYD, hydrocarbon; N, nitrogen; OM, organic matter; P, phosphorous; PW, pore water; SEM, scanning electron microscope; S&C, silt and clay.

Taxonomy

Light micrographs of preserved specimens were made with a Zeiss Axioplan (body) and a Zeiss Stemi 2000–c (chaetae) stereomicroscopes equipped with a CMEX5 digital camera (Euromex). For scanning electron microscope (SEM) observations, the worms were prepared using standard SEM procedures as described in Martin et al. (2003).

Type and non-type specimens of the new species are deposited in the Biological Collections of the Centre d’Estudis Avançats de Blanes (CEAB, Blanes, Catalunya, Spain), the Museo Nacional de Ciencias Naturales (MNCN, Madrid, Spain), Museu Nacional de História Natural e da Ciência (MB, Lisboa, Portugal), National Museum Wales (NMW, Cardiff, Wales, UK) and Senckenberg Research Institute and Natural History Museum (SMF, Frankfurt, Germany). The specimens of S. parva are deposited in the CEAB and the MNCN.

The electronic version of this article in Portable Document Format (PDF) will represent a published work according to the International Commission on Zoological Nomenclature (International Commission on Zoological Nomenclature, 1999), and hence the new names contained in the electronic version are effectively published under that Code from the electronic edition alone. This published work and the nomenclatural acts it contains have been registered in ZooBank, the online registration system for the ICZN. The ZooBank LSIDs (Life Science Identifiers) can be resolved and the associated information viewed through any standard web browser by appending the LSID to the prefix http://zoobank.org/. The LSID for this publication is: urn:lsid:zoobank.org:pub:0FCB04BD-5ACA-4808-8091-67F7EF106021. The online version of this work is archived and available from the following digital repositories: PeerJ, PubMed Central SCIE, DigitalCSIC and CLOCKSS.

Morphometry

The number of chaetigers was used as surrogate of size. All relevant morphological features susceptible to be size-dependent were identified and counted or measured in a selected subset of 21 entire individuals of the new species of Sigambra (Table S2). Dry weight was estimated after drying (60 °C) for 24 h. The relationships between size and morphological features were assessed by Pearson correlation, using the same procedures and software as for the ecological analyses.

Results

Ecology

Neither the most relevant environmental descriptors, DDP and HYD, nor the populations of the targeted polychaetes show significant differences during the three surveys (Tables S3A and S3B). Thus, for the purposes of this study, all surveys are considered together.

When analyzing the distributions of the target species with respect to the sediment descriptors (Table S3C), the most common pattern is a significant increase with rising levels of Ba, occurring for all species except S. parva and P. trionyx. The new species of Sigambra, L. sebastiena, Raricirrus sp. and Ampharetidae sp. also show significant increasing densities with rising levels of HYD. Among them, the first two show the same trend in relation with OM, but the density of L. sebastiena decreases with the rise of PW. Finally, both the densities of Capitella sp. and Ampharetidae sp. increase with rising levels of N and decrease with increasing levels of CS and P, with those of Capitella sp. also increasing with S&C and those of Ampharetidae sp. significantly decreasing with DDP.

The first axis of the PCA (Eigenvalue 15.5, 84.4% of the explained variation) is defined by the opposite trend of Ba and, especially HYD, on the negative (left) sector and DDP on the positive (upper) sector (Fig. 2). Axis two (Eigenvalue 2.15, 11.7% of the explained variation) is defined by the opposite trend of S&C on the negative (lower) sector and, especially CS, on the positive (upper) sector (Fig. 2). The densities of the new species of Sigambra and S. parva plotted on the PCA axis clearly show their disjunct distributions, highlighting the particularly high densities of the former in the stations with HES (Fig. 2).

Figure 2 Results of the PCA based on environmental parameters.

Samples grouped according to groups I, II and III; species densities (ind/m2) plotted as grey circles. DDP, distance from drilling point; HYD, total hydrocarbons; Ba, barium; CS, coarse sand; S&C, silt and clay; OM, organic matter; PW, pore water; N, nitrogen; P, phosphorous; Snk, Sigambra nkossa sp. nov.; Sp, S. parva; Csp, Capitella sp.; Pt, Paramphinome trionyx; Rsp, Raricirrus sp.; Ob, Oxydromus berrisfordi; Ls, Lindaspio sebastiena; Amp, Ampharetidae sp.

The sampling stations cluster in three significantly different groups (ANOSIM: Global R = 0.846, significance level = 0.1%) according to the PCA, which coincide with markedly different HYD ranges (Fig. 2). The significance of the differences is confirmed by PMDA (Table S4B). Group I includes the stations 5 and 6 (HYD >20,000 mg/kg DW), group II includes the stations 4, 7, 13, and 18 (HYD: 20,000–100 mg/kg DW) and group III includes the stations 1–3, 8–11, and 15–16 (HYD < 100 mg/kg DW). The most significant differences occur always when group III is involved (Table S4A). Group I is very homogeneous but, within groups II and III, two subgroups can be respectively distinguished. Characteristically, the smallest subgroups including stations 4, 13 and 18 (from 2003) and 13 (from 2002) in group II, and 2, 13, and 19 (all from 2003) in group III, are mainly isolated from the other stations due to the extremely low CS (<2%). The differences between these two subgroups and the remaining stations of each group prove to be also significant (ANOSIM: Global R = 0.944/0.630, significance level = 2.9%/1.5%, respectively).

When analyzed individually, DDP, HYD, and Ba show highly significant differences between the PCA groups, but also S&C and P slightly differ (Table S4D, Fig. 3). The pairwise analyses (Table S5) show that: in group III, DDP is always significantly higher and HYD and Ba are always significantly lower; neither DDP, nor Ba differ between groups I and II, while HYD is significantly much higher in the former; S&C is only slightly lower in group I than in group II, while none of them differ from group III, and there is no clear pattern for P (Fig. 3). Finally, CS, PW, OM and N do not show significant differences between the PCA groups (Table S5, Fig. 3).

Figure 3 Environmental descriptors in the N’Kossa field, plotted against the groups I, II and III obtained in the PCA.

DDP, Distance from drilling points (m); HYD, Total hydrocarbons (as g/kg of sediment). Ba, Barium (mg/kg of sediment). CS, coarse sand (%). S & C, silt and clay (%). PW, pore water (%). OM, organic matter (%). N, nitrogen (%). P, phosphorous (%).

Figure 4 Sigambra nkossa sp. nov. light microscopy images.

(A) Whole body, dorsal view. (B) Anterior end, dorsal view. (C) Anterior end, ventral view. (D) Detail of chaetigers 2 (lacking ventral cirrus) and 3, ventral view. (E) Everted proboscis, dorsal view. (F) Everted proboscis, ventral view. (G) Everted proboscis, lateral view. (H) Parapodium 3. (I) Midbody parapodium. (J) Detail of hook (right) and spine (left) from I. (K) Posterior most parapodium. (L) Detail of hook (right) and spine (left) from K.

The density of the target benthic species also shows significant differences in the PCA groups (Table S4C), with all species having also clearly different distribution patterns (Table S5, Fig. 2). The new species of Sigambra, Capitella sp. and O. berrisfordi have non-significantly different densities between groups I and II, which are significantly higher than those in group III. In fact, the new species of Sigambra and O. berrisfordi are completely absent from group III, as it is L. sebastiena. Raricirrus sp. shows significantly higher densities in group I than in group III, while those in group II do not differ from those in groups I and III. Paramphinome trionyx shows non-significant differences in density between group I and groups II and III, while the density is significantly higher in group II than in group III. Ampharetidae sp. shows significantly higher densities in group II than in groups I and III, which, do not differ from each other. Finally, Sigambra parva is completely absent form group I, and shows similar low densities in groups II and III.

Taxonomic account

Family Pilargidae Saint-Joseph, 1899	
Sigambra Müller, 1858	

Sigambra. Müller (1858): 214 (original description); Pettibone (1966): 179 (genus reinstated with key to species); Licher & Westheide (1997): 2 (key to species and synoptic table with characters of all species); Nishi et al. (2007): 65 (synoptic table with characters of all species); Salazar-Vallejo et al. (2019): 24 (diagnosis), 44–46 (key to species).

Type species. Sigambra grubii Müller, 1858, by monotypy.

Diagnosis. Based on Glasby & Salazar-Vallejo (2022). Pilarginae with body depressed, usually obconic. Prostomium with three antennae, longer than palps; palps biarticulate. Tentacular cirri as long as half width of tentacular segment. Parapodia biramous. Dorsal and ventral cirri foliose to tapered, dorsal ones usually longer than ventral ones. Notopodium with one strongly recurved dorsal hook, or with one strongly recurved dorsal hook and one slightly bent spine along many chaetigers, sometimes with capillary chaetae or with protruding tips of notoaciculae. Neuropodia with shorter pectinate capillaries, medium-sized denticulate capillaries, and longer finely denticulate capillaries, often twisted distally.

Remarks. Virtually all morphological features characterizing Sigambra have been well defined in recent literature (Salazar-Vallejo et al., 2019; Glasby & Salazar-Vallejo, 2022). Our observations on the species found in the N’Kossa field, complemented with the original descriptions and figures of other species of the genus (Table S6), lead us to suggest that the generic diagnosis probably requires to be amended in what concerns the so called “capillary chaetae” associated with the notopodial hooks/spines. In our specimens, these supposed “capillary chaetae” revealed to be the tips of the notoaciculae protruding from the acicular lobes. Accordingly, the notopodial character “sometimes with accessory capillaries” may need to be replaced by “sometimes with protruding tips of notoaciculae”. However, further examination of other species having these “capillary chaetae” is required prior to confirm the necessity of amending the generic diagnosis. While waiting for this review, we have included in the diagnosis “sometimes with capillary chaetae or with protruding tips of notoaciculae”, instead of just “sometimes with capillary chaetae”.

Sigambra nkossa sp. nov.	
LISID: urn:lsid:zoobank.org:act:9015B0CC-4C6B-4DFF-A9AE-F403F512AE2B.	
Figs. 4–5	

Material examined: Holotype: CEAB-A.P. 913A, entire specimen, N’Kossa oil and gas field, St. 13(2), Republic of Congo, Gulf of Guinea, Atlantic coasts of western Africa (approximately 05°16′S, 11°34′E), collected by DM in April 2003, on soft sediments at approximately 180 m depth, fixed in a 4% formaldehyde/seawater solution, preserved in 70% ethanol. Paratypes: CEAB-A.P. 913B, 1 specimen in two fragments, anterior end dissected to reveal the pharynx, other data as in holotype; CEAB-A.P. 913C, 1 entire specimen, prepared for SEM, collected in March 2002, St. 13.2, other data as in holotype; CEAB-A.P. 913D: 245 specimens, collected in March 2002, St. 6.2, other data as in holotype; CEAB-A.P. 913E: 173 specimens, collected in March 2002, St. 4.3, other data as in holotype; CEAB-A.P. 913F: 153 specimens, collected in March 2002, St. 6.3, other data as in holotype; CEAB-A.P. 913G: 431 specimens, collected in March 2002, St. 13.2, other data as in holotype; CEAB-A.P. 913H: 267 specimens, collected in April 2003, St. 4.3, other data as in holotype; CEAB-A.P. 913I: 749 specimens, collected in April 2003, St. 6.1, other data as in holotype; CEAB-A.P. 913J: 479 specimens, data as in holotype; MNCN 16.01/19138: 10 specimens, data as in holotype; MB29-000359–MB29-000368: 10 specimens, data as in holotype; NMW.Z.2022.002.0001: 10 specimens, data as in holotype; SMF 31812–31821: 10 specimens, data as in holotype. Additional specimens are available from the authors.

Diagnosis. Prostomium bilobed, with biarticulate palps; median antenna ca. 1.7 times longer than lateral ones; proboscis with fourteen pointed distal and numerous subdistal papillae; peristomium with a transverse row of epidermal papillae; dorsal tentacular cirri ca. 1.8 times longer than ventral ones, dorsal parapodial cirri 1.5 to three times longer than ventral cirri (which are absent in chaetiger 2); pointed neurochaetae serrated and pectinated; notopodial hooks from chaetiger 5 to body end, notopodial spines from chaetiger 9 to body end and tips of notoacicula protruding from acicular lobe from mid-body parapodia to body end.

Description. Body measuring 2.8–26 mm long, 0.6–1.9 mm wide (with parapodia) for 23–134 chaetigers. Body dorsoventrally flattened, with faint blackish dorsal and ventral pigmentation on prostomium, peristomium, antennae and tentacular cirri, almost disappearing in preserved specimens (Figs. 4A–4C). Eyes lacking (Figs. 4A and 4B). Prostomium bilobed; palps biarticulated, with large palpophores and small palpostyles (Figs. 5A, 5C and 5D); interpalpal area distinct, anteriorly depressed, widely expanded posteriorly (Figs. 4B and 5C). Three antennae slender, tapering distally, on posterior half of prostomium; median antenna 1.5 to 2.0 (1.7 on average) times longer than lateral ones; lateral antennae clearly surpassing palp tips, median antenna reaching chaetiger 2 when folded backwards (Fig. 4A and 4B); lateral antennal furrow distinct, slightly divergent (Fig. 5A).

Tentacular segment three times wider than longer, with two pairs of slender and subequal tentacular cirri, ventral one as long as lateral antennae, dorsal cirri ca. 1.8 times longer than ventral cirri; two small anteriorly projected lobes between lateral antennae and tentacular cirri and a central anteriorly projected lobe ventrally (Figs. 4B, 4C, 5A and 5D). Small epidermal papillae, 5–9 µm in diameter, near posterior prostomial margin, in a single row laterally, in several rows medially, between bases of lateral antennae (Figs. 5A and 5B). Rows of similar epidermal papillae on posterior margin of each parapodium, dorso-laterally near bases, 11–16 papillae on each side.

Proboscis with fourteen distal conical papillae, with long pointed ends in two lateral groups of four papillae, usually bigger than dorsal (three papillae) and ventral (three papillae) groups (Figs. 4F, 5C and 5D); curved surface with two central longitudinal rows with numerous randomly distributed small papillae on both sides dorsally (Fig. 4E), numerous randomly distributed small papillae ventrally (Fig. 4F) and five bigger papillae on a transversal row and several randomly distributed and progressively smaller papillae laterally (Figs. 4G and 5D); all papillae pointed, with cusps directed backwards in totally everted proboscis.

Parapodia sesquiramous; dorsal and ventral cirri tapering distally; first dorsal cirri 1.7 times longer than central antenna, 1.6 times longer than dorsal tentacular cirri, 2.5 times longer than remaining dorsal cirri; following dorsal cirri 1.5 longer than ventral cirri in anterior chaetigers, becoming progressively longer and slender in middle and posterior chaetigers, to up to three times longer in posterior most chaetigers; second chaetiger with dorsal cirri shorter than following ones, lacking ventral cirri (Figs. 4A–4D, 4H, 4I, 4K). Notopodia with dorsal cirrus and one or two pointed aciculae, one with curved end protruding from acicular lobe in posterior chaetigers and one straight, non-protruding; one emergent hook from chaetiger 5 (5–6), one emergent spine from chaetiger 9 (9–12), both present until posterior most chaetigers, more exposed posteriorly (Figs. 4I–4L, 5G, 5H). Neuropodia well-developed; ventral cirri similarly shaped but shorter than dorsal ones, extending far beyond neuropodial lobe tips, longer posteriorly (Figs. 4A, 4B and 4D); acicula straight, pointed, sometimes slightly protruding outside acicular lobe; parapodial lobe blunt, almost square; about 10 short pectinate supraacicular chaetae with long spinulation and filiform tips; about same number of slightly longer, distally pointed pectinate chaetae; numerous minutely serrated, distally pointed, capillary chaetae of variable length but much longer than the other neurochaetae; spinulation always directed upwards, thicker in shorter chaetae and in proximal part of longer chaetae (Figs. 5E and 5F).

Pygidium with two slender anal cirri, pointed, of about same length as first dorsal cirri (Fig. 4A).

Remarks. Sigambra nkossa sp. nov. perfectly fits all diagnostic characters of the genus, except for the presence of spines in addition to hooks and protruding tips of notoacicula in most notopodia along the body, a peculiar feature also described for Sigambra robusta (Ehlers, 1908), Sigambra bassi (Hartman, 1947) and Sigambra healyae Gagaev, 2008 (Ehlers, 1908; Hartman, 1947; Licher & Westheide, 1997; Gagaev, 2008) that allows distinguishing them well from all other species of the genus (Table S6). The presence or absence of stout emergent notochaetae and, when present, whether they are hooked or straight, were considered to be diagnostic characters at generic level within Pilargidae (Pettibone, 1966; Blake, 1997; Gil, 2011). However, we agree with Glasby & Salazar-Vallejo (2022) in that the presence of stout emergent notopodial chaetae must prevail over the fact that these may be represented by both hooks and spines in at least part of the notopodia along the body in the generic diagnosis. The origin of these spines and their relationships with the notopodial hooks and aciculae needs to be further investigated, but their singularity seems to indicate an apomorphic character.

Other species of Sigambra were previously found in the same region of western Africa. Among them, S. robusta was collected by the German Valdivia Expedition in the muddy sediments of the “Große Fisch-Bucht” (10–12 October 1898, Baía dos Tigres or Tigres Bay, southern Angola) (Chun, 1903; Ehlers, 1908), an area geographically close to the offshore Congo location where we found S. nkossa sp. nov. Tigres Bay was formerly delimited westwards by a long sandy peninsula running parallel to the coast and connected to mainland at its southern region by an isthmus; besides, it had muddy sediments that were organically enriched in some parts (Chun, 1903). The isthmus was disrupted by the ocean in 1962 giving rise to Tigres Island, which is separated from mainland by the Tigres Strait. This changed the prevailing ecological conditions of the type locality of S. robusta, which have been under the direct influence of the local coastal currents since that time.

Sigambra robusta was reported in Valdivia stations 76 and 77 (Ehlers, 1908), with the station 76 being the type locality according to Licher & Westheide (1997). While the depth of these stations was not referred in the original description, it was mentioned as being “ca. 14 m” for station 76, in the collection data of Nereis lucipeta Ehlers, 1908. However, this last species was based on epitokous males collected in mass on the surface of the water during the night, attracted by lights, as mentioned by Ehlers (1908) on page 71: “Die beigelegte Etikette enthält die Angabe: ‘In Masse an der Oberfläche des Wassers, unter den Lampen” [“The enclosed label contains the indication: “In mass at the surface of the water, under the lamps”.”]. Thus, the associated station depth seems to be an estimate, as it was not mentioned elsewhere in the publication. Still, both the official contemporary charts of the bay (Commissão de Cartographia, 1896; Hydrographic Office, 1915) and the official expedition account (Chun, 1903) reported the overall maximum depth in the collection area (i.e., the northern region of Tigres Bay), as being around 20 m depth. The species was posteriorly recorded from other regions of Angola (including the exclave of Cabinda, just 70 km away from N’Kossa) and Namibia, between 20 and 150 m depth (Kirkegaard, 1983; Licher & Westheide, 1997), and there is a possible report from Senegal at 23 m (Augener, 1918).

Indeed, although the lower bathymetric limit of S. robusta reaches the depth of the bottoms around the N’Kossa platforms, its recorded habitat is overall around ten times shallower than that of S. nkossa sp. nov. Moreover, the new species shows pigmentation in the anterior end (absent in S. robusta), lacks the second ventral cirri (present in S. robusta) and has notopodial hooks from chaetiger 5–6 (instead of 43–70 in S. robusta).

Figure 5 Sigambra nkossa sp. nov. SEM images.

(A) Anterior end, dorsal view. (B) Detail of the anterior end, dorsal view. (C) Anterior end with everted proboscis, frontal view. (D) Anterior end with everted proboscis, lateral view. (E, F) Neuropodial chaetae from mid-body. (G) First segments with notopodial hook and spine (white arrows pointing on the first two spines). (H) Notopodial hook and spine from mid-body (white arrow pointing on protruding tip of notoacicula).

Figure 6 Sigambra parva. SEM images.

(A) Anterior end, lateral view. (B) Anterior end, ventral view. (C) Detail of anterior end, frontal view. (D) Detail of anterior end, dorsal view. (E) Anterior end with everted proboscis, lateral view. (F) Posterior end.

Figure 7 Sigambra parva. SEM images.

(A) Mid-body parapodia, dorsal view. (B) Detail of hooks from mid-body parapodia. (C, D) Neuropodial chaetae from mid-body.

Sigambra nkossa sp. nov. differs from S. healyae from the Arctic Ocean in having faint blackish dorsal and ventral pigmentation on prostomium, peristomium, antennae and tentacular cirri (instead of red spots on prostomium in S. healyae), notopodial hooks starting from chaetiger 5–6 (instead of 4 as in S. healyae), notopodial spines from chaetiger 9–11 (instead of 12–15 in S. healyae) and neuropodial capillaries present (absent in S. healyae), as well as in being reported from different geographical locations and depths (off Congo, 180 m depth vs. Canadian Basin, 1,800 m depth).

Sigambra bassi was originally described from Florida (Gulf of Mexico) and reported repeatedly since then as having notopodial hooks together with emergent spines (Pettibone, 1966; Wolf, 1984; Blake, 1997; Licher & Westheide, 1997; Moreira & Parapar, 2002). However, in its original extended description, Hartman (1947) clearly distinguishes the acicula from the notopodial spines, while discussing the presence of structures such as “notoacicular spines [whose] free end is strongly curved”, “dorsal acicular spines”, “recurved acicular spines” or “projecting hooks”. Moreover, the author illustrated a parapodium with an emerging straight spine that seems to be the prolongation of the acicula, being thus difficult to ascertain if spines were really present along with hooks, or if this was referring to the emerging tips of the acicula. This question was clarified by Pettibone (1966), who revised type and non-type material from Florida and stated “stout notopodial hooked notoseta beginning about setiger 14 (11–15); occasionally additional single emergent notoseta, straight or slightly curved (called an aciculum by Hartman)”, while representing the notopodial hooks and spines as structures clearly independent from the notoacicula, originating from a different and much more superficial region (Pettibone, 1966, Fig. 16), just as it occurs in S. nkossa sp. nov.

With just a few exceptions (e.g., Pettibone, 1966; Wolf, 1984), most available descriptions of S. bassi, including the original one, comprehended material from both Atlantic and Pacific USA coasts (e.g., Hartman, 1947; Licher & Westheide, 1997) or just from the Pacific coasts (Blake, 1997). This mixture of different populations, representing very likely different taxa, explain certainly the wide range of registered character variation, such as the starting of notopodial hooks between chaetigers 3–25 (Licher & Westheide, 1997). This drawback was previously referred by other authors, and work is now under progress to solve it, likely implying erecting new taxa (see Salazar-Vallejo et al., 2019 and references therein). This fact led us to compare S. nkossa sp. nov. only with descriptions from the Gulf of Mexico, discarding both those based on (or including) the Pacific specimens referred above and those based on American Atlantic populations recorded outside the Gulf of Mexico (e.g., Gardiner, 1976). Accordingly, N. nkossa sp. nov. can be distinguished by the presence of notopodial hooks from chaetigers 4–6, instead of 10–15 in S. bassi and, also, by having a median antenna reaching only chaetiger 2 when bent backwards, instead of chaetiger 12 in S. bassi (but see comments on this last character in Discussion).

Etymology. The specific epithet “nkossa” is considered as a substantive in apposition, referring to the N’Kossa oil and gas field, off the Republic of Congo, the type locality of the species.

Distribution. Known only from the N’Kossa oil and gas field (approximately 05°16′S, 11°34′E), western Atlantic Ocean, about 60 km off the coasts of the Republic of Congo (Gulf of Guinea); associated with HES (Fig. 1).

Sigambra parva Day, 1963	
Figs. 6–7	

Material examined. All specimens collected in the N’Kossa oil and gas field, Republic of Congo, Gulf of Guinea, Atlantic coasts of western Africa (approximately 05°16′S, 11°34′E), by ED and DM, on soft sediments at around 180 m depth, fixed in a 4% formaldehyde/seawater solution, preserved in 70% ethanol. CEAB-A.P. 922A, one specimen, collected in November 2000, St. 1; CEAB-A.P. 922B, one specimen, collected in March 2002, st. 2.6; CEAB-A.P. 922C, two specimens, collected in March 2002, st. 3.1; CEAB-A.P. 922D, three specimens, collected in March 2002, st. 3.1; CEAB-A.P. 922E, one specimen, collected in March 2002, st. 2.1; CEAB-A.P. 922F, two specimens, collected in April 2003, st. 2.1; CEAB-A.P. 922G, four specimens, collected in March 2002, st. 4.3; CEAB-A.P. 922H, four specimens, collected in March 2002, st. 15.3; CEAB-A.P. 922I, two specimens, collected in March 2002, st. 16.1; CEAB-A.P. 922J, three specimens, collected in March 2002, st. 16.2; CEAB-A.P. 922K, four specimens, collected in April 2003, st. 18.1; CEAB-A.P. 922L, seven specimens, collected in April 2003, st. 18.2; CEAB-A.P. 922M, one specimen, collected in April 2003, st. 18.3; MNCN 16.01/19139, four specimens collected in April 2003, st. 18.2.

Description of specimens from the study area. Body dorsoventrally flattened, tapering toward pygidium. Prostomium bilobed, with palps fused at basis, biarticulate, with large palpophores and small palpostyles (Figs. 6A–6C and 6E); with three slender antennae on posterior prostomium region, tapering distally (Figs. 6A and 6C); median antenna about 1.5 times as long as lateral ones; four subdermal eyes in trapezoidal arrangement, sometimes non-visible. Peristomium with two pairs of slender tentacular cirri, subequal or with dorsal one slightly longer, and longer than lateral antennae (Figs. 6A–6B); two small anteriorly projected lobes between lateral antennae and tentacular cirri (Figs. 6A and 6D). Small epidermal papillae dorsally on posterior prostomium margin (Figs. 6A and 6D), and on posterior dorsolateral margin of each chaetiger and near parapodial bases. Fourteen distal proboscideal papillae, conical with round tips (laterally, one pair on each side, big; two groups of five small papillae each, one ventral and one dorsal) (Figs. 6A–6B and 6E); curved surface with two subdistal rows of oval, non-pointed papillae; three triangular, pointed papillae, one median and two on each lateral behind subdistal rows, with cusps pointing backwards (Fig. 6E). Parapodia sesquiramous. Notopodia with 1–2 aciculae and pointed, slender dorsal cirrus. First chaetiger with dorsal cirri longer than tentacular cirri and following dorsal cirri; second chaetiger with much shorter dorsal cirri; hooks from chaetiger 4–5, more strongly recurved and emergent posteriorly (Figs. 7A–7B). Neuropodia well-developed, with conical acicular lobes, one straight acicula and ventral cirri similar to dorsal ones but shorter, extending beyond tip of acicular lobes and absent in chaetiger 2 (Fig. 6B); numerous pointed, minutely serrated simple chaetae (Fig. 7C); 1–6 short pointed pectinate supracicular chaetae on posterior chaetigers (Fig. 7D). Anal cirri slender, as long as dorsal tentacular cirri (Fig. 6F).

Remarks. The morphology of the specimens of S. parva found in the N’Kossa field agrees overall with that of the types redescribed by Moreira & Parapar (2002) from Cape Province (South Africa), except in the number of supracicular pectinate neurochaetae, which may be up to 6 in the Congolese specimens and were reported as being 1–2 in the type specimens. Some Congolese specimens showed four subdermal eyes, which were not visible in others, while the apparent lack of eyes in the South African specimens may be attributed either to the fading effect of their long storage in preservation fluid or to the hiding by a thicker epidermal layer.

There seem to be some slight differences (mainly concerning the distribution of the proboscideal papillae) between the Congolese specimens and both the types of S. parva and the specimens from the Iberian Peninsula (Columbretes, NW Mediterranean; Baiona, NE Atlantic), identified as S. parva by Moreira & Parapar (2002). However, we have not been able to either confirm or reject this statement in light of our own observations, which were inconclusive. Considering the great distances separating the different populations, it is plausible to think that further work based on larger sets of specimens and, ideally, combining additional morphological observations with morphometric and molecular analyses, will help in assessing whether these and other populations considered as belonging to S. parva from areas far from the type locality may represent or not different taxa.

Distribution. Originally described from the south coast of Cape Province, South Africa (34°10′S, 23°32′E), where it occurred down to 100 m depth on muddy bottoms (Day, 1963), and later reported from the Atlantic and Mediterranean coasts of the Iberian Peninsula between 2 and 122 m depth in muds and shelly and fine sands with seagrass meadows (Moreira & Parapar, 2002), in the Aegean Sea (Arvanitidis, 2000; Faulwetter et al., 2017), in the western coast of South Africa (Kirkegaard, 1983) and in the N’Kossa field off the Republic of Congo (this article).

Morphometric analysis of S. nkossa sp. nov.

Most morphological characters measured are size-dependent (Pearson correlation > 0.7), particularly dry weight, width of chaetiger 15 (with parapodia), length of median antennae, and length of the first dorsal and ventral cirri (Pearson correlation > 0.9) (Table S7, Fig. 8). Conversely, only a few characters do not vary with size, most of them being ratios between different anterior appendages (antennae and tentacular dorsal and ventral cirri) and only three (i.e., the starting chaetiger of notopodial hooks and spines, and the absence of ventral cirri on chaetiger 2) refer to characters previously considered as species-specific (Table S6).

Figure 8 Sigambra nkossa sp. nov. selected morphological features and their respective relationships with size, expressed as number of chaetigers.

(A) Total length (µm). (B) Dry weight (mg). (C) Width of chaetiger 15 (with parapodia, µm). (D) Length of median antennae (µm). (E) Length of first dorsal cirri (µm). (F) Starting segment for the dorsal capillary chaetae. (G) Length ratio for the antennae (middle/lateral). (H) Length ratio for the tentacular cirri (dorsal/ventral). (I) Starting segment for the notopodial hook. A–F, size-dependent characters; G–I, size-independent characters.

Discussion

Ecology

The first benthic samples here analyzed were collected one year after the impact on the sediments caused by oily drill cuttings. Drilling muds were mostly deposited in the surrounding bottoms just below the two N’Kossa platforms and their amount was clearly decreasing with the increasing DDP, as previously reported by Denoyelle et al. (2010). This pattern was highly consistent during all three surveys, to the extent that, with a few exceptions, the environmental parameters analyzed did not differ with time, in a similar way as it happened with the density distribution patterns of the population of the polychaete species targeted in this article.

As a result, the stations located at a greater distance from the platforms (PCA group III) were clearly isolated from the impacted ones (PCA groups I and II), as proven by the significant presence in the latter of Ba. In all three surveys, the stations were grouped in one or another of the two impacted groups independently, to some extent, from DDP. This was caused by the local dominant currents that generated different levels of accumulation whose distribution depended not only on DDP, but also on the location around the platforms. In turn, our statistical analyses showed very small differences in granulometry, except for the slight gradient observed in the second PCA axis, which is likely a residual effect of the drilling deposits causing the presence of slightly bigger particles in the most HYD enriched stations (Dalmazzone et al., 2004; Denoyelle et al., 2010). This clearly supported that, before the impact, the analyzed stations had all the same type of sedimentary bottoms. Nevertheless, the drilling disposal caused the HYD concentration to be much higher in group I than in group II stations. Therefore, the main reason explaining the differences in the distribution patterns observed for the polychaete species here targeted was the presence of HYD, as it also occurred for the whole macrofaunal community (Denoyelle et al., 2010).

Slope and shelf seeps represent organic-rich areas in the overall poor deep-sea bottoms (Washburn, Demopoulos & Montagna, 2018), where they may function as local, ephemeral disturbances supporting infaunal species pre-adapted to organic-rich, reducing environments (Levin et al., 2000). Natural HYD seeps provide unique habitats to many different species, although macrofaunal organisms often receive less attention than megafauna and microorganisms (Levin, 2005). However, it has been suggested that, at higher taxonomic levels, there was no specific infauna that may be considered as seepage indicators (Washburn, Demopoulos & Montagna, 2018). Conversely, at the species level, there are numerous well-known inhabitants of these particular environments, such as siboglinid tubeworms (Schulze & Halanych, 2003; Bright & Lallier, 2010; Karaseva et al., 2020) and other polychaete species belonging to Orbiniidae (Blake, 2000), Chrysopetalidae (Aguado, Nygren & Rouse, 2013), Dorvilleidae (Ravara, Wiklund & Cunha, 2021) and Hesionidae (Desbruyères & Toulmond, 1998). Among hesionids, for instance, Sirsoe methanicola (Desbruyères & Toulmond, 1998) was originally described from methane seeps (Desbruyères & Toulmond, 1998) and later found also inhabiting fossil fuel reserves (Fisher et al., 2000).

Sigambra nkossa sp. nov. responded positively to the presence of Ba in the sediments, indicating a connection with drilling cuttings, but the relationship was stronger with respect to the effective presence of HES. Specimens of Sigambra were already present in a survey carried out in 1994, just at the starting of the drilling activities. However, their identity cannot be confirmed and there are no materials available to be revised (JM Amouroux, C Labrune, pers. comm., 2022). This Sigambra showed densities from 3 to 30 ind/m2 and hardly tolerated HES (E. Dutrieux 1996, unpublished data), which agrees with the 2000–2003 pattern of S. parva (up to 50 ind/m2), rather than with that of S. nkossa sp. nov. (from 1,000 ind/m2 in 2000 to more than 4,000 ind/m2 in 2003). We thus suggest that: (1) the specimens of 1994 could belong to S. parva, whose presence followed the typical composition of the local benthic assemblages in non-disturbed bottoms and, although it tolerated HYD to some extent, it did not proliferate in HES, and (2) that S. nkossa sp. nov. appeared in the study area in connection with the presence of HES, with its bloom (and thus its close relationship with HYD) being certainly unique among the currently known species of Pilargidae.

Similar to S. nkossa sp. nov., all other polychaete species targeted in this article were present in HES and virtually absent (or showed very low densities) in non-impacted sediments. The distribution patterns of the whole macrofaunal assemblage were suggested to be equivalent to those traditionally associated with eutrophication phenomena in areas impacted by waste disposals (Denoyelle et al., 2010). However, only S. nkossa sp. nov. and L. sebastiena responded positively to an increase in OM, and only Capitella sp. showed increasing densities in correlation with increasing S&C. In turn, all studied species responded positively to Ba (except P. trionyx and, obviously, S. parva), while S. nkossa sp. nov., Raricirrus sp., L. sebastiena, and Ampharetidae sp. also responded specifically to HYD.

Among the described species of Raricirrus, R. maculatus Hartman, 1961 was found as living in highly organically enriched sediments off California, while R. beryli Petersen & George, 1991 was locally abundant in sediments from northern North Sea oil fields (Hartman, 1961; Petersen & George, 1991). Lindaspio sebastiena is, at present, known only from the N’Kossa field. Curiously enough, despite being relatively highly abundant in 2000 and 2002, it almost disappeared in 2003, in parallel with the enormous increase in abundance of S. nkossa sp. nov. exactly in the same bottoms. Therefore, as suggested by Dalmazzone et al. (2004), we cannot discard a cause relationship between these two trends, with the competence with the pilargiid (which was effectively able to reproduce in the area) explaining the rapid decrease in abundance of the spionid (which reproduction in the area has not been observed).

The relationship with HES in the N’Kossa field was not restricted to polychaetes, as the large vesicomyid bivalve I. bigoti was originally described there (Cosel, & Salas, 2001) and later found in cold seeps and methane hydrate areas off Gabon and in the Congo basin (Rodrigues et al., 2012). This species was misidentified as either Anodontia cf. edentula or Loripes cf. contrarius by Denoyelle et al. (2010). It was very abundant (>500 ind/m2) in the same HES sediments inhabited by S. nkossa sp. nov. and the other HES associated polychaetes studied here, and was completely absent from bottoms devoid of HYD. The depth in the N’Kossa region (around 150 m depth) is unusually shallow for vesicomyids and its presence was initially explained as a result of the high OM concentration caused by the Congo River discharges (Cosel & Salas, 2001). The low salinities associated with the Congo River plume were detectable on the water surface even at 200 km offshore (Vangriesheim et al., 2009), which included the waters surrounding the N’Kossa platforms (i.e., about 60 km offshore). This was also perceptible by the recurrent presence of small “floating islands” formed by living fragments of riverine vegetation, sometimes with animals on them (D Martin, pers. obs., 2003). However, the riverine influence was not specifically concentrated in the immediate vicinity of the platforms. Moreover, according to our data, the unusually high OM concentrations underneath the platforms were only related to drilling mud disposals and, thus, to the presence of HES. Taking into account that Vesicomyidae include well-known inhabitants of chemosynthetic environments like hydrocarbon seeps (Audzijonyte et al., 2012), we strongly suggest that the occurrence of I. bigoti below the N’Kossa platforms, rather than being caused to the riverine influence, was instead following the presence of HES, as reported for other vesicomyids in the region (Cosel & Olu, 2009).

Deep-sea hydrocarbon seeps occur all around the world oceans (Sibuet & Olu, 1998; Levin, 2005) and are particularly abundant in the West African oceanic region (e.g., Olu et al., 2009; Warén & Bouchet, 2009; Jatiault et al., 2018). Therefore, we suggest that some of the species here studied and, particularly, the polychaetes S. nkossa sp. nov., L. sebastiena and Raricirrus sp., and the bivalve I. bigoti, may be native inhabitants of natural HES that colonized alternatively the unusual shallow depths around the N’Kossa platforms thanks to the HYD artificial source provided by drilling muds. Consequently, better than an opportunistic response associated with an anthropogenic disturbance, its presence in the artificial HES around these platforms could be an expression of its normal mode of life, with the human driven offshore HYD routine exploration and extraction activities acting as stepping stones facilitating dispersal and colonization of these somewhat atypical shallow bottoms to species natively associated with deeper environments having natural HES.

Morphometry

In addition to the rise of modern molecular techniques, morphometric analyses have also shown to be a valid and trustful method to assess morphological discrimination between cryptic polychaete species (Koh & Bhaud, 2001; Koh & Bhaud, 2003; Ford & Hutchings, 2005; Glasby & Glasby, 2006; Lattig, San Martín & Martin, 2007; Martin et al., 2017). Morphometry is particularly useful when some of the relevant characters routinely used to define the species within a genus are quantitative and may be, to some extent, size-related. In traditional approaches, ranges are often provided, but this usually depends on the author, so that they are not available for all affected species and relevant characters, as can be easily perceived for the known species of Sigambra (Table S6).

Moreover, morphometric observations may be also used to determine whether a single measurement from a single individual may or may not be taxonomically relevant by assessing the variability at the population level. Based on a series of specimens representative of the whole size range of the population of S. nkossa sp. nov., we demonstrate that most quantitative characters commonly used to define species in the genus Sigambra are strongly size-dependent and show a highly significant positive correlation with size. This correlation may explain more than 70% of their variability, or even more than 90% in the case of dry weight, width of chaetiger 15 (with parapodia), length of median antenna, and length of the first dorsal and ventral cirri. Therefore, these characters should be avoided when diagnosing a species, unless their variability at the population level can be reported and, if possible, statistically compared to those of the closely related species.

Conversely, a robust taxonomic relevancy can be attributed to size-independent characters. These include the length ratios for the main anterior appendages (i.e., the dorsal tentacular cirri being much longer than the ventral ones, and the dorsal parapodial cirri being much longer than the ventral ones), as well as the only three discrete or binary features referring to specific morphologic characters (i.e., in the case of S. nkossa sp. nov., the starting of the notopodial hooks at chaetiger 5–6, the starting of the notopodial spines at chaetiger 9–12, and the absence of the ventral cirri at chaetiger 2), which are consistently constant all along the size range of the studied population. Thus, these characters can be considered unambiguous at the species level and species-specific at the individual level.

In the particular case of the presence of hooks, the starting chaetiger has been recently reported as being either invariable (or vary for a very few chaetigers, more or less at random) or as starting more posteriorly in larger specimens (Salazar-Vallejo et al., 2019). Therefore, care should also be taken when attributing size-independence to a given character, as this may also vary significantly according to the species and along its ontogeny.

Conclusions

Our results allowed us to suggest that native inhabitants of natural deep-sea HES may alternatively colonize unusual shallow depths around oil and gas platforms thanks to the HYD artificial sources provided by drilling mud deposits. Off the Republic of Congo, in the N’Kossa oil and gas field, the polychaetes L. sebastiena, Raricirrus sp. and S. nkossa sp. nov., together with the bivalve I. bigoti, may be examples of this circumstance. Therefore, their association with artificial HES would be more an expression of their normal mode of life, rather than just an anthropogenically triggered opportunistic response.

Accordingly, the offshore oil and gas routine exploration and extraction activities could act as stepping stones for the dispersal of benthic inhabitants native from different natural deep-sea environments with HES.

The current knowledge of the species of Pilargidae, and particularly those belonging to Sigambra, seems to be incorrect regarding the presence of the “capillary chaetae” accompanying the notopodial hooks and spines. According to our observations, at least in S. nkossa sp. nov. these structures are, in fact, protruding tips of notoacicula. However, examining other species having these “capillary chaetae” is required prior to decide whether the generic diagnosis should be amended by replacing the notopodial character “sometimes with accessory capillaries” by “sometimes with protruding tips of notoaciculae”.

Moreover, generic diagnostic characters such as “presence/absence of stout emergent notochaetae” or “stout notochaetae either as hooks or as spines” seem to be in conflict with the definition of some of the current known genera of Pilargidae, as well as with most of the identification keys available for the family (e.g., Pettibone, 1966; Blake, 1997; Gil, 2011). If these identification keys are followed strictly, species having both types of stout notochaetae (straight and hooked) on the same notopodial bundle (such as S. robusta, S. bassi, S. healyae and S. nkossa sp. nov.) would be keyed as probably belonging to a new genus. Therefore, while waiting for further studies (including molecular analyses) helping to clarify the taxonomic weight of the presence and type of these notopodial spines, we consider the character “presence of stout emergent notochaetae” as prevailing over “stout notochaetae either as hooks or as spines”. Also, we agree with Glasby & Salazar-Vallejo (2022) in that the diagnosis of Sigambra must include the character “notopodium with one strongly recurved dorsal hook or with one strongly recurved dorsal hook and one slightly bent spine along many segments”. As so, forthcoming identification keys of the family should state “stout emergent notochaetae straight, not hooked” vs. “stout emergent notochaetae also including hooked chaetae”, instead of “stout emergent notochaetae straight, not hooked” vs. “stout emergent notochaetae hooked”.

Finally, our morphometric analyses allowed us to demonstrate that most quantitative characters commonly used to define the species of Sigambra are strongly size-dependent, proving that they must be avoided for species definition, except when it is possible to define statistically their intrapopulation variability to be compared with those of the most closely related species. Conversely, size-independent characters must be considered as taxonomically robust and unambiguous at the species level, as well as species-specific at the individual level.

Supplemental Information

Table S1 Abundance of the targeted polychaete species and main environmental descriptors during the three sampling years at the N’Kossa oil and gas field

Snk: Sigambra nkossa sp. nov.; Sp: Sigambra parva; Csp: Capitella sp. Pt: Paramphinome trionyx. Rsp: Raricirrus sp. Ob: Oxydromus berrisfordi. Ls: Lindaspio sebastiena. Amp: Ampharetidae sp.; DDP: distance from drilling point (m); HYD: Total hydrocarbons (mg/kg of sediment); Ba: Barium (mg/kg of sediment); CS: coarse sand (%); S&C: silt and clay (%); PW: pore water (%); OM: organic matter (%); N: Nitrogen (%); P: Phosphorous (%).

Click here for additional data file.

Table S2 List of morphological measurements used in the morphometric analyses of Sigambra nkossa sp. nov

DC: dorsal cirri; VC: ventral ciri.

Click here for additional data file.

Table S3 Relationships between species abundance and environment descriptors

(A) Results of the Parametric Multidimensional analysis for the environment descriptors. (B) Results of the Parametric Multidimensional analysis for the species abundance according to the sampling years. (C) Results of the Pearson correlation analysis between species abundance and environment descriptors. Significant differences/correlations are highlighted in bold. MD: Mahalanobis distance; Pearson: correlation coefficient; p: significance level; DDP: Distance from drilling points; HYD: Total hydrocarbons; Ba: Barium; CS: Coarse sand; S&C: Silt and clay; PW: Pore water; OM: Organic matter; N: Nitrogen; P: Phosphorous.

Click here for additional data file.

Table S4 Results of the analyses for the environment descriptors and species abundances according to the PCA station groups

(A) ANOSIM pairwise tests (environmental descriptors). (B) Results of the Parametric Multidimensional Analysis for the sediment descriptors. (C) Results of the Parametric Multidimensional Analysis for the density of target species. (D) Results of the independent one-way ANOVAs for each sediment descriptor and target species. (E) Pairwise post-hoc comparisons (Tuckey test). Bold: Significant differences. NS: Non significant; R: R statistic; PP: Possible permutations; AP: Actual permutations; N: Number ≥ Observed; DDP: Distance from drilling point; HYD: Total hydrocarbons; Ba: Barium;CS: Coarse sand; S&C: Silt and clay; PW: Pore water; N: Nitrogen; P: Phosphorous; DF: degrees of freedom; SS: Sum of Squares; MS: Mean squares; F: Fisher’s test; p: significance level.

Click here for additional data file.

Table S5 Results of the Tuckey HSD pairwise comparisons for the environmental descriptors and the target species densities, according to the PCA station groups

D: difference; SD: Standard difference; CV: Critical value; P: probability; DDP: distance from drilling point; HYD: Total hydrocarbons; Ba: Barium; CS: coarse sand; S&C: silt and clay; OM: organic matter; PW: pore water; N: Nitrogen; P: Phosphorous; Snk: Sigambra nkossa sp. nov.; Sp: S. parva; Csp: Capitella sp. Pt: Paramphinome trionyx. Rsp: Raricirrus sp. Ob: Oxydromus berrisfordi. Ls: Lindaspio sebastiena. Amp: Ampharetidae sp.

Click here for additional data file.

Table S6 Summary of the main morphological characters for the currently known species of Sigambra Müller, 1858.

DC, dorsal cirri; LA, lateral antennae; MA, median antenna; TC, tentacular cirri; VC, ventral cirri; +/-, presence/absence of the character; ?, not reported.

Click here for additional data file.

Table S7 Summary of the measurements for the main morphological characters of Sigambra nkossa sp. nov. ( N = 21), grouped according to the significance of their correlation with size (as number of chaetigers)

A. Size-dependent. B. Not varying with size. AVG, average; STD, standard deviation; Min, minimum; Max, maximum; Pearson, correlation coefficient; P, significance level; D, dorsal; V, ventral.

Click here for additional data file.

We would like to thank TotalEnergies E&P Congo for partly sponsoring the field surveys and for their help with the sampling campaigns, the staff of the N’Kossa platform for their kind welcome and help during our visits; the staff of the Laboratoire Municipal et Régional de la Ville de Rouen (France) for the analyses of the sediment characteristics; José Manuel Fortuño from the SEM Service of the CMIMA-CSIC (Spain), for his help with SEM micrographs; Alexandra Rizzo and an anonymous reviewer for their detailed revisions of the manuscript; and the curators of the different biological collections where the specimens examined were deposited, for their assistance.

Additional Information and Declarations

Competing Interests

Author Contributions

Field Study Permissions

Data Availability

New Species Registration

Sébastien Thorin, Romain Le Gall & Éric Dutrieux are employed by CREOCEAN. Claude-Henri Chaineau is employed by TotalEnergies.

Daniel Martin conceived and designed the experiments, performed the experiments, analyzed the data, prepared figures and/or tables, authored or reviewed drafts of the article, and approved the final draft.

João Gil performed the experiments, analyzed the data, authored or reviewed drafts of the article, and approved the final draft.

Claude-Henri Chaineau conceived and designed the experiments, authored or reviewed drafts of the article, and approved the final draft.

Sébastien Thorin performed the experiments, authored or reviewed drafts of the article, and approved the final draft.

Romain Le Gall performed the experiments, authored or reviewed drafts of the article, and approved the final draft.

Eric Dutrieux conceived and designed the experiments, performed the experiments, authored or reviewed drafts of the article, and approved the final draft.

The following information was supplied relating to field study approvals (i.e., approving body and any reference numbers):

Field studies were aproved by TotalEnergies E&P Congo.

The following information was supplied regarding data availability:

Abundances of polychaetes and values of environmental descriptors used in the ecologial analyses, a list of the morphological measurements used in morphometry and four tables summarizing the main statistics associated to the results are available in the Supplemental Files.

The following information was supplied regarding the registration of a newly described species:

Sigambra nkossa LSID: urn:lsid:zoobank.org:act:9015B0CC-4C6B-4DFF-A9AE-F403F512AE2B.

Publication LSID: urn:lsid:zoobank.org:pub:0FCB04BD-5ACA-4808-8091-67F7EF106021.

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
