# Peer review of "Description of the new species Sigambra nkossa (Annelida, Pilargidae), with an analysis of the distribution patterns of polychaetes associated with artificially hydrocarbon-enriched bottoms"

_PeerJ, doi:10.7717/peerj.13942_

## Round 0.1 · original submission · Minor Revisions

Dear authors,

Many thanks for your submission to PeerJ. Your manuscript was reviewed by two specialists who have made several suggestions as you will see in the attached files. One reviewer indicated that it needed Major Revision because of the numerous alterations it needs whereas a second reviewer indicated that it needed Minor Revision. As you may see, both reviewers and I agree that an emendation to the generic diagnosis without examining the type species should be reconsidered. Looking forward to receiving your revised manuscript.

Reviewer 1 ·

Basic reporting

I am not a native English speaker, but I think that the article is technically correct and has professional standards of courtesy and expression.
This paper reports description of a new species and, the results of distribution of eight polychaete species found in association with sediments enriched with hydrocarbons as a result of the gas and oil drilling activities around the N’Kossa field, Republic of Congo. It is an interesting work carried out in a geographic area and in a scientific issue few explored, which has the ‘standard sections’ used in the journal, however, there are some observation that should be clarified:
- The title does not mirror the information displayed in the manuscript, it prioritizes the distribution patterns of polychaetes (which are 8 species), but, e.g., in the abstract this point includes only 2 rows, while the information about the new species has 13 rows; or in conclusions, where 10 rows are linked to distribution of the 8 polychaete species, but the conclusions on the new species has 32 rows. So, I think that the title must be focused on description of the new species and secondarily on the distribution of the 8-target species.
- The introduction is focused on the study area and it would be extremely useful that the authors provide information on similar areas in other geographic regions around of hydrocarbons fields, where studies on polychaetes have been carried out.
- The information in the taxonomic section can be improved, since the support to erect the new species, or the “extended diagnosis” in the case of S. parva, are found out of the taxonomic section, in “discussion”, which pause the arguments to establish the new species. So, a point of “remarks” placed later of “description” could be better. I suggest that authors follow the format used by Tovar-Hernández et al. (2020, PeerJ 8:e9692 DOI 10.7717/peerj.9692) to explain the taxonomy of polychaete species, which could improve the taxonomic explanation.
In the “Material examined”, it is necessary to include the complete date of the collected material; also, information as “several hundred of specimens collected in 2000, 2002 and 2003…” are not adequate to posterior reviews of these material. To include the catalogue numbers of the type material deposited in the museums.
In S. parva, the concept of “extended diagnosis”, could be incorrectly used, since the authors did not review the material type or indicated other material examined, and notes as “with four subdermal eyes in trapezoidal arrangement”, which is not indicated in the original description or in other revisions, could create confusion on its interpretation. So, I think that a headline “description of specimens from the study area” could be more appropriate.
The figure 3 is not relevant, when this figure is referred in results, its not contributed with substantial information; it could be to information showed in figure 2.
The figure 2 has not sufficient resolution, and the labels in the “figure-Snk” are indistinguishable (PCA-vectors).
The information of Table 3 cannot be reviewed, it must correspond to “Tuckey HSD pairwise comparisons”, but the data belongs to morphological characteristics of Loimia species (¡¡).
In all supplementary material is necessary to include the used abbreviations, the headlines of table are confused.

Experimental design

As it was previously indicated, the study should clearly define that the taxonomic examination of species is its main aim.
In general, the methodology is consistent with the collection and analysis of data and the research was conducted in conformity with the prevailing ethical standards in the field. However, it is necessary to describe in detail some data on the sampling stations and the implemented test to support the results:
- The “enriched sediments (HES)” are a synonymous of “HYD concentrations”?, or how the HES was measurement, it is confuse along the text.
- How the “replications” (3) were used in the data analysis?
- How and why the eight target polychaete species were selected?
- How many “sediment descriptors” were evaluated, how they were measurement, and why some of these were overlooked or not examined in the manuscript. In the “supplementary_data_1” file, the abbreviations of the descriptor names are not explained and they cannot be identified.
- The depth values are not relevant in this study, why?
- The parametric multidimensional analysis based on the Mahalanobis distance has been usually used to determine the similitude between multidimensional variables, so its use to evaluate the temporal and spatial differences in the sediment descriptors and in the occurence of the eight polychaete species must be explained in detail as an alternative to this analysys type in future works. It is important to indicate the hypothesis testing in this Mahalanobis method. Here, as in all test, it is also necessary to show the alfa-value used.
- Why a log-transformation of data was used?, and how you evaluated the assumption of normality and homoscedasticity?, can you include these results?
- The results of the Tuckey pairwise comparisons are not in the table 3 or in the manuscript.
- Can you include the museum´s code where the specimens were deposited?
- If you have the total length of specimens, why you are using the number of chaetigers and no the length to determine the size-depending characters ?, explain.
- The rates between lenght of antena and peristomial cirri, can hide their lineal relationships and to bias the conclusiones as size-independet characters?

Validity of the findings

- Sampling design should be reviewed, I recommend to include in the ecological analysis the factor “platform”, since samples from the NFK1 and NKF2 were studied, but they are not examined in the statistical test. So, also other variables as “location around the platforms”, depth, “effective presence of HES)”, which affect the distribution of species, as the authors indicated in the discussion section, could be also included in the analysis.
- The discussion is center on information previously found in the study area, but it would be very important to include information on areas affected by oil activities on other geographic regions, or reviewing studies in these environments examining the effect of variables, e.g. barium, on the polychaete occurrence.
- The discussion and conclusions are focused in the taxonomic information, so I think that the erection of the new species is valid and their description is appropriated. The observations about the morphological variations of S. parva are adequate, but the previously observations about the “extended diagnosis” must be kept in mind.
- The examined data are provided as supplementary material; however, they must be formatted to explain the abbreviation used.
- The conclusions are focused on the taxonomic information, so the authors must reorganize the manuscript and their aims to highlight the description of the new species and the effect of some environmental variables on its distribution. The notes on the “addition to identification keys of the family Pilargidae” is confused and inappropriate in the context of the study.

Additional comments

This is an interesting work carried out in a geographic area and in a scientific issue few explored, however, the previous comments should be review in detail before to accept its publication.

·

Basic reporting

Comments and suggestions can be found throughout the text.

Experimental design

Comments and suggestions can be found throughout the text.

Validity of the findings

Comments and suggestions can be found throughout the attached file.

---

## Round 0.2 · Minor Revisions

Dear authors,

Many thanks for taking in all of the reviewers' suggestions, especially those concerning the generic emendation. Please see some additional minor changes required concerning the sampling years and stations.
Looking forward to receiving the final version of your manuscript.

Best wishes,
Wagner

Reviewer 1 ·

Basic reporting

I think that the changes made in the manuscript correctly cover the previous suggestions made along the manuscript sections, and the modifications carried out in the taxonomic issue, ordering some paragraphs in the taxonomy, results and remarks sections, clarify and significantly support the erection of the new species.
The author decisions on the figures, tables and supplementary material are also suitable.
However, it is necessary to clear some inconsistencies related to sampling years and their associated stations:

- In methods, it was indicated that sites from 2000, 2002 and 2003 were sampled, but in the ecological analysis were examined samples from 2000, 2001 and 2002. Explain these discrepancies.

- In table 1, sites from 2000, 2001 and 2002 are referred, but in results, the obtained station groups, according to cluster analysis, say: “…the smallest subgroups including stations 4, 13 and 18 (from 2003) and 13 (from 2002) in group II, and 2, 13, and 19 (all from 2003) in group III…”. So, these discrepancies follow, it is necessary to correct the sampling years.

- In this sense, in the ecology results say: “Neither the most relevant environmental descriptors, DDP and HYD), nor the populations of the targeted polychaetes show significant differences during the three surveys (Supplementary tables 3A, 3B)”, but in those tables only comparisons between 2001 and 2002 were showed; and the comparisons with data of 2000 (or 2003?)?

- In table 4C, there are not comparison with group I.

Experimental design

No comment.

Validity of the findings

No comment.

Additional comments

No comment.

---

## Round 0.3 · accepted · Accept

Dear authors,

Many thanks for agreeing with the reviewer's suggestions.